# Bulk Polymerization of Acrylic Acid Using Dielectric-Barrier Discharge Plasma in a Mesoporous Material

**DOI:** 10.3390/polym15132965

**Published:** 2023-07-06

**Authors:** Matthew Mieles, Sky Harper, Hai-Feng Ji

**Affiliations:** Department of Chemistry, Drexel University, Philadelphia, PA 19104, USA

**Keywords:** DBD plasma, non-thermal plasma, mesoporous materials, poly(acrylic acid)

## Abstract

This research investigated a non-thermal, dielectric-barrier discharge (DBD) plasma-based approach to prepare poly(acrylic acid) (PAA) from acrylic acid in its liquid state at atmospheric temperature and pressure. Neither additives nor solvents were needed, and the polymerization was accomplished both as a film and inside a sheet of mesoporous paper. All prepared samples were characterized and the DBD plasma-initiated kinetics were analyzed for the polymerization of acrylic acid. Using FTIR semi-quantitative analysis, the degree of polymerization was monitored, and the reaction followed an overall second-order kinetic model with respect to the DBD-initiated polymerization. Additionally, the application of a PAA-modified paper as a water retention cloth or ‘wet wipe’ was investigated. The results showed that the PAA-modified paper substrates using DBD plasma increased water retention as a function of plasma treatment time.

## 1. Introduction

Poly(acrylic acid) (PAA) is a pH-responsive addition polymer with various applications, such as in hydrogels [1], superabsorbent materials [2], drug delivery [3], artificial tears to treat dry eye syndrome [4], surface modifications for cell adhesion and proliferation [5], and as a mucoadhesive gel coating [6]. PAA is biocompatible, biodegradable, and has low toxicity in human blood [7,8]. PAA is usually formed via a free-radical polymerization mechanism and initiated by conventional UV and chemical methods. These polymerizations are usually conducted in aqueous or organic solvents and require several reagents or additives for a quality polymerization to occur. For example, nitroxide is required in an acidic medium for the nitroxide-mediated polymerization of acrylic acid (AA) [9]. In atom transfer radical-polymerization (ATRP) of AA, typical ATRP reagents are needed [10]. The same is the case for reversible addition−fragmentation transfer (RAFT), where common chain transfer agents are needed [11]. Only a few works have demonstrated the bulk polymerization of AA and even then the use of initiators cannot be avoided [12]. These initiators, due to their inherent properties, produce radicals which can be very dangerous reactive species for many applications where PAA is suitable. The removal of the unreacted initiators in many cases can be troublesome and even harder to quantify. For these reasons, methods for polymerizing monomers without the need for any of these toxic chemicals are needed.

Dielectric-Barrier Discharge (DBD) plasma is a less commonly used method for polymerization and is ideal for producing thin polymer films to coat many surfaces. DBD atmospheric plasma uses ambient air to produce an ionized gas composed of charged and free radical species at room temperature and atmospheric pressure. This is achieved by applying a high voltage between two electrodes [13]. To prevent electron flow between these electrodes, a dielectric material is applied to at least one of them. The insulator prevents a build-up of high currents between the electrodes, creating electrically safe plasma without substantial gas heating. Consequently, DBD plasma is considered safer to utilize and is commonly referred to as “cold plasma” in contrast to high-temperature plasmas [14,15]. Notably, DBD devices can be designed as handheld devices, offering convenient operation similar to handheld UV devices.

Research on the applications of discharged plasma is currently thriving in various fields, including biomedical devices, environmental science, and agriculture [16,17]. Among the wide range of plasma techniques, DBD plasma stands out due to its exceptional safety and selectivity when interacting with biological systems [18,19]. The effects of DBD plasma treatment encompasses numerous beneficial outcomes such as being antimicrobial and utilized for the sterilizatiofwn of skin [20], facilitating wound healing [21], sterilizing root canals [22], enhancing cell transfection [23], and promoting cell proliferation [24]. In order to advance the understanding of the interactions between plasma and living cells and tissues, and to facilitate the clinical applications of plasma, extensive research has been devoted to the investigation of chemical species generated during plasma treatments. The majority of studies investigating plasma treatments have focused on samples in aqueous solutions, where the organic chemicals present include sugars, lipids, and amino acids, among others. These organic compounds are significant constituents of a cell culture medium. In these studies, it was observed that the organic chemicals underwent decomposition, leading to the formation of smaller chemical species. However, recent investigations have revealed a new and intriguing phenomenon when treating sugar powders, such as ribose and glucose, in the solid phase. Contrary to the expected decomposition, these sugars, particularly ribose, underwent polymerization. In following studies, DBD was also used to initiate the bulk polymerizations of bithiophene [25] and aminophenol [26] inside a piece of paper without any other chemicals added. 

One of the major advantages of DBD plasma is that it does not require any initiators, nor chemical additives for polymerization, which allows for the bulk polymerization of many gas, liquid, and solid-state monomers. There are also instances when UV and heat cannot be used to polymerize certain heat- and UV-sensitive materials, nor can radiation efficiently polymerize a monomer where direct radiation cannot occur, such as internally to a material or a material that has a low radiation penetration depth. The air plasma consists of reactive oxygen and nitrogen species. These species possess the capability to initiate polymerization reactions due to their ability to penetrate thin monomer films and/or mesoporous materials. The DBD plasma polymerization of AA has been studied using other plasma generation setups, and the operating parameter effects on the polymer film product have been discerned [27]. Several properties of the PAA films produced using DBD plasma have also been investigated, such as the homogeneity and hydrophilic characteristics [28]. However, in all of the works previously conducted, the polymerization of AA using plasma-generating devices has come from depositing PAA onto a substrate from AA in its gaseous state, and usually mixed with some other carrier gas to facilitate the generation of plasma and free radicals [29,30,31]. 

Herein, AA was polymerized and studied from its liquid state using DBD plasma to produce thin films of the polymer. Considering the distinct nature of the presented plasma-based approach for synthesizing PAA films on a wide range of substrates, this method was termed as DBD non-gaseous polymerization. This nomenclature serves to differentiate this approach from the conventional gaseous plasma polymerization method. In comparison to other polymerization methods, non-gaseous DBD plasma polymerization is a favored method for polymerization in mesoporous materials since atmospheric-pressure plasma has been seen to penetrate planar mesoporous substrates with thicknesses similar to paper and even in thin wood planks (1–2 mm in thickness) [32]. This internal polymerization in thicker substrates is mostly due to the sample preparation, where the mesoporous substrate to be polymerized is soaked and effectively impregnated with the liquid monomer and then subsequently polymerized by plasma initiation. For some low-pressure plasmas where the monomer is deposited from its gas phase and on to non-planar substrates, this penetration depth is much lower (less than 1 mm) [33]. In this work, the characterization and kinetics were studied for the bulk polymerization of AA using DBD plasma without any additives or solvents added, and it was demonstrated that liquid-phase AA can indeed be bulk polymerized as a film and in a piece of paper via the DBD plasma process at atmospheric conditions. The plasma treated AA-modified paper samples showed improved water retention properties relative to the untreated samples. This work will provide valuable insights into the DBD non-gaseous plasma polymerization of PAA, which may be used in industrial applications due to its lower cost, short treatment times, and free of solvents and additives. 

## 2. Experiments and Materials

### 2.1. Materials

Acrylic acid (>98% Beantown Chemical, Hudson, NH, USA) was used as received. Poly(acrylic acid) (MW ~ 1,000,000, Polysciences, Inc., Warrington, PA, USA). Letter paper (Office Depot ImagePrint, Boca Raton, FL, USA, 77–89% cellulose, uncoated).

### 2.2. DBD Plasma Generation

The DBD air plasma was generated by using a microsecond-pulsed power supply (FID Technology, Burbach, Germany) and an electrode dielectric-barrier discharge setup as shown in Figure 1. The DBD electrode works by creating a plasma stream between a high voltage 25 mm thick copper plate and the ground. A 1 mm thick quartz dielectric plate is used as an insulating barrier to cover the copper plate. The plasma discharge gap between the bottom of the quartz plate and the surface of the samples was kept constant at 5 mm. The plasma is generated by using a variable voltage and variable frequency power supply that applies a pulsed alternating polarity voltage of 20 kV (peak to peak) with a 10 ns pulse width and a rise time of 5 V/ns. For all the experiments, a peak voltage of 11.2 kV and repetition frequency of 690 fHz were used. The input energy used was calculated to be around 10 mJ/pulse. The working area of plasma treatment was equivalent to the copper plate dimensions of 38 mm × 64 mm.

### 2.3. DBD Plasma Polymerization of AA

A thin film of AA was applied on a 1 cm × 1 cm gold-coated silicon substrate by drop casting 10 μL of the monomer using an autopipette. This yielded a liquid AA film of approximately 100 μm in thickness. The gold substrates were used for their IR inactivity, which allowed the AA bands to be discerned along the plasma treatment process without any IR band interference from the substrate. The AA-coated substrates were then treated with DBD plasma under ambient conditions (normal temperature and pressure), and the relative humidity fluctuated day-by-day between 60 and 70%. Three AA-coated substrates were then treated with DBD plasma under atmospheric conditions for 30 s, 1, 2, 3 and 5 min intervals.

### 2.4. FTIR Characterization and Analysis

A PerkinElmer Spectrum One FT-IR Spectrometer (Waltham, MA, USA) was used to obtain the Fourier-transform infrared (FTIR) spectra of the samples before and after plasma polymerization. FTIR sampling was performed by attenuated total reflection (ATR) over the range 650 cm^−1^ to 4000 cm^−1^ with a resolution of 4 cm^−1^. The background FTIR spectra were collected on a clean gold substrate for the thin films and in air for the AA-modified samples. OriginLab was used for deconvolution and quantification of IR peaks to analyze the spectra at the various time intervals of DBD plasma treatment. 

### 2.5. Wet Wipe Application

The 15 mm × 10 mm paper samples were modified with AA by allowing them to soak overnight in 20 mg of the monomer. This was enough AA to fully submerge the entire sample. The samples were removed from the soaking liquid and dabbed off with a paper towel to remove excess AA on the surface. Several samples were then subjected to the same plasma treatment times of 1, 2, and 5 min. After the plasma treatment, 100 µL of distilled water was auto pipetted on to each sample and the weight of the samples was taken every five minutes over the course of one hour. All masses were measured on a pre-calibrated analytical balance with 0.1 mg resolution. These results were compared to a paper with just monomer (not polymerized) and pristine paper. The environmental conditions were controlled and kept constant during this time period, with a room temperature of 23 °C and a relative humidity of 53%.

## 3. Results and Discussion

### 3.1. Characterization of AA and PAA Using Infrared Spectroscopy

Figure 2 shows the infrared spectrum of AA and a reference PAA sample. Based on previously reported IR spectra of AA and PAA [34], the polymerization of AA to PAA upon exposure to DBD plasma could be validated. The vibrational band located at 1700 cm^−1^, which is attributed to the stretching mode of the carbonyl bond, was observed throughout all of the samples, regardless of plasma treatment time—as expected. The other peaks of interest are displayed in the table inset of Figure 2 and are in reference to the vinylidene group of AA. The band at 1636 cm^−1^ for the C=C out-of-phase stretching mode is observed to completely disappear in the FTIR spectrum of PAA. Many of the other vibrational modes are seen to be suppressed in PAA relative to AA but only the 1636 cm^−1^ band fully disappears, as is expected in a fully polymerized AA sample where no more C=C bonds remain. This 1636 cm^−1^ band that should decrease with respect to the degree of polymerization was used in the kinetic study to monitor the ability of DBD plasma to polymerize AA films. 

### 3.2. Kinetic Study

The carbonyl band was used as an internal standard reference peak to determine the degree of polymerization of AA as a function of DBD plasma treatment time. The IR transmittance data were converted to absorbance data and the peak areas were calculated for the IR bands at 1636 cm^−1^ and 1700 cm^−1^. Using OriginLab, deconvolution for the peaks of interest was accomplished during the various plasma treatment times. The peak area ratio of the 1636 cm^−1^ (A_1_) band to that of the 1700 cm^−1^ (A_0_) band was calculated for each plasma treatment time, and this value was plotted against the treatment time to display the degree of polymerization. As can be seen in Figure 3a, the peak area ratio (A_1_/A_0_) of these two bands were used to semi-quantitatively demonstrate the successful polymerization of AA by DBD plasma. 

The semi-quantitative analysis method for determining the reaction rate order using FTIR was adapted from Pintar et al. [35] and was used in a previous paper to demonstrate the degree of conversion of PEGDA using DBD plasma [32]. A second-order treatment of the data yielded the best linear fit, with an R-squared value of 0.98218 (Figure 3b). This implies that the overall order of the plasma-initiated polymerization of AA is second order. Equivalent results were seen in two previous works using DBD plasma, which may imply that DBD plasma polymerization for bulk and solid-state polymerizations follows a second-order kinetic mechanism.

### 3.3. Wet Wipe Application

One major advantage of the DBD plasma bulk polymerization method over other methods is the ability to initiate a polymerization inside a mesoporous material, such as inside a piece of printing paper. A paper substrate was soaked in acrylic acid and subjected to plasma for several time intervals. Figure 4 shows the FTIR spectra of an AA-modified paper sample before and after 5 min of plasma exposure. The band at 1636 cm^−1^ decreases significantly relative to the 1700 cm ^−1^ band after 5 min of plasma exposure, indicating a high degree of polymerization.

Due to the hydrophilic properties of PAA, one potential application of the PAA-modified paper is its use as a wet wipe that retains water longer than a pristine sheet of paper. The water retention property of the paper was assessed by monitoring its weight loss over time and the results are displayed in Figure 5.

To determine the rate of water loss in each sample, a linear fit was applied to each set of data, and the slope values gave an approximation of the water loss over time. The rate of water loss was found to be almost identical for the pristine paper and AA-modified paper. For the pristine paper, the water loss rate was calculated to be 2.8 mg/min, and the rate was 2.9 mg/min for the AA-modified paper. However, for the plasma-treated samples, the rate decreased by more than half in all cases. The samples subjected to plasma for 1, 2, and 5 min were found to lose water at a rate of 1.4, 1.3, and 0.9 mg/min, respectively. Water retention increased as a function of plasma treatment time.

## 4. Conclusions

The characterization and kinetics for the polymerization of liquid-phase acrylic acid using a non-thermal dielectric-barrier discharge plasma method without the use of any additives or solvents has been demonstrated. The acrylic acid can also be polymerized inside a piece of paper using the DBD plasma method and the water retention property of the plasma-treated paper samples showed that water retention increases as a function of plasma treatment time. The successful DBD polymerization of liquid AA at room temperature and using atmospheric conditions allows for the production of any polymer film directly from its monomer liquid state. The polymer films can be formed on virtually any planar substrate using a very simple plasma-generating device as the one demonstrated. This makes the plasma polymerization of AA more straightforward and possible without the need of complex setups as used in gas-phase plasma polymerization.

In comparison to other polymerization methods, atmospheric-pressure plasma is unique in its ability to penetrate mesoporous materials, such as paper, and facilitate polymerization of the monomers within. It was shown that using AA and other vinyl monomers (in previous works), it is possible to polymerize monomers internally, which allots different substrates new and/or improved properties relative to the pristine samples. Using other polymerization methods requires the use of toxic initiators and additives that in applications such as wet wipes can leave behind unreacted chemical residues that are hard to remove post polymerization. DBD plasma polymerization is a favorable method for bulk polymerization in these porous environments and provides a greener and safer approach. In future work, a parametric study will be conducted to determine the effect of the plasma discharge conditions on the properties of the polymer films produced from liquid-state monomers.

## Figures and Tables

**Figure 1 polymers-15-02965-f001:**
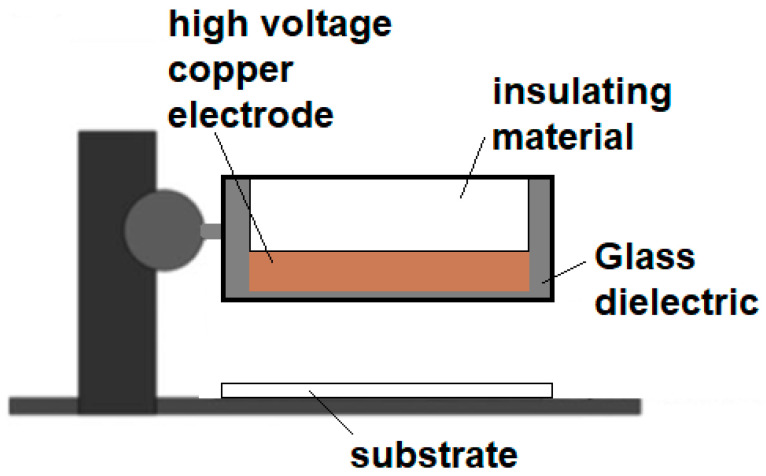
Scheme for the DBD plasma device.

**Figure 2 polymers-15-02965-f002:**
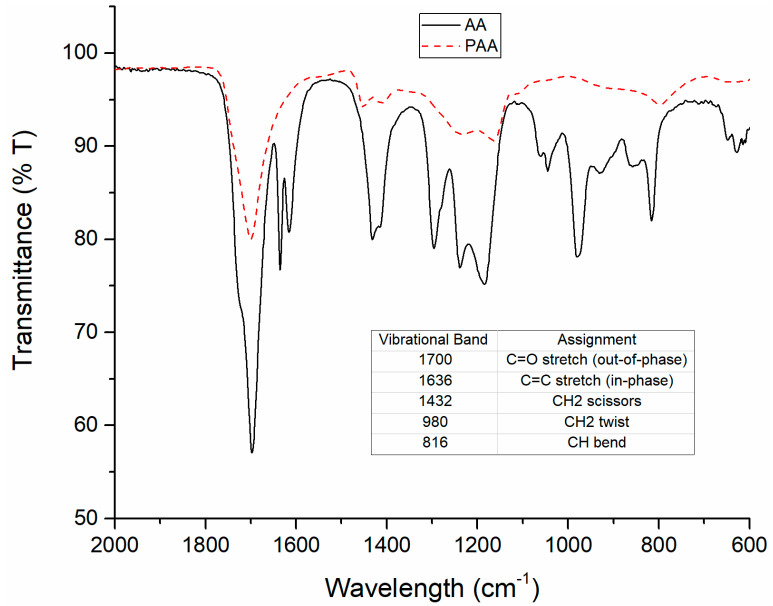
FTIR spectra of acrylic acid (AA) and polyacrylic acid (PAA). Inset shows IR vibrational assignments for the carbonyl and vinylidene groups of AA.

**Figure 3 polymers-15-02965-f003:**
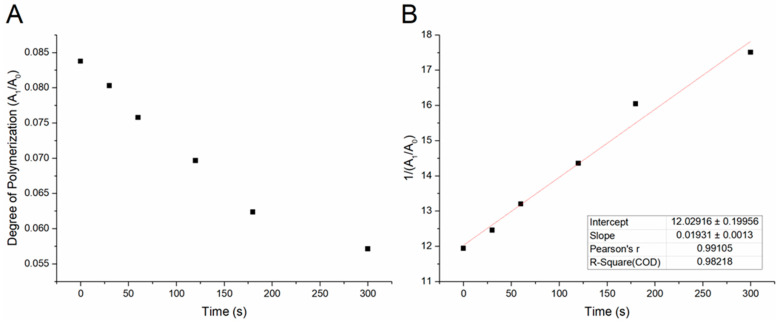
(**A**) Plot of the FTIR peak area ratios (A_1_/A_0_) as a function of plasma treatment time. The A_1_ and A_0_ denote the peak areas of the 1636 cm^−1^ and 1700 cm^−1^ vibrational modes, respectively. (**B**) A second-order treatment of the A_1_/A_0_ ratio was plotted by graphing the inverse of this ratio as a function of treatment time. The inset shows several relevant parameters for the linear fit with an R-squared value of 0.98218.

**Figure 4 polymers-15-02965-f004:**
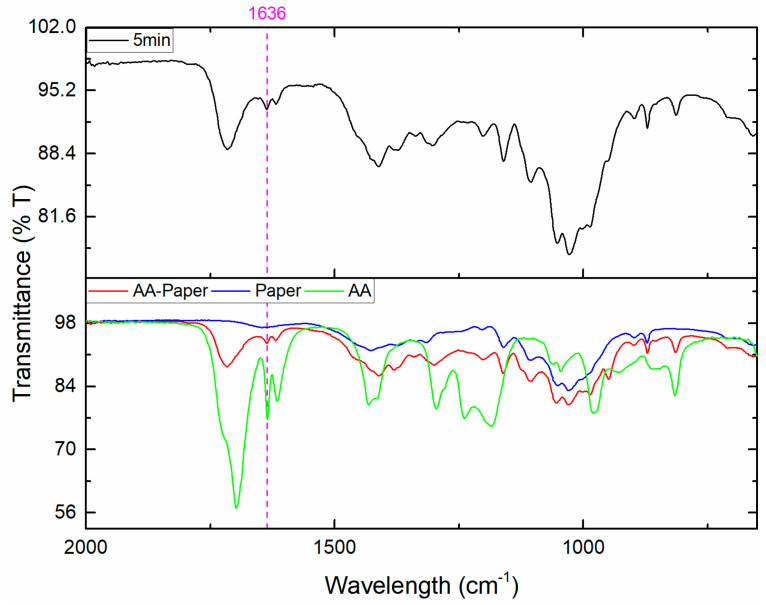
FTIR spectra of acrylic acid monomer and paper substrate samples before and after 5 min of plasma exposure.

**Figure 5 polymers-15-02965-f005:**
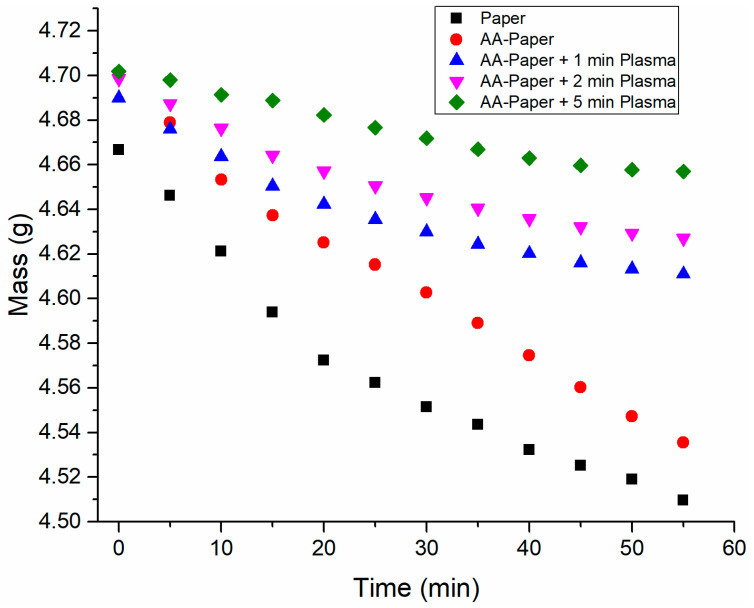
The mass loss due to water as a function of time. The black squares represent pristine paper, red circles represent the AA-modified paper before plasma treatment, and the blue triangles, magenta triangles, and green rhombuses represent the AA-modified paper after plasma treatments of 1, 2, and 5 min, respectively.

## Data Availability

Not applicable.

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
