# Peer review of "Bulk Polymerization of Acrylic Acid Using Dielectric-Barrier Discharge Plasma in a Mesoporous Material"

_polymers, 2023, doi:10.3390/polym15132965_

Round 1

Reviewer 1 Report (Previous Reviewer 2)

I highly appreciate the efforts of the authors to improve the quality of the manuscript. I think that they have taken into account most of my comments. Nevertheless, there are still some points that I would like to remark before the article gets published.

I find that the conceptualization of the study should be improved from a plasma point of view. Polymerization of Acrylic Acid (AA) using non-thermal plasmas is not a novel technique. The authors removed a reference they cited in the previous version (https://doi.org/10.1002/ppap.200800089) that could be valuable to be cited. As it is mentioned in this revisited article, plasma polymerization of AA has been studied for several decades and some comments about that could be included. Throughout the manuscript, the authors insist on the fact that the polymerization is carried out without additives or solvents. I completely understand and support this idea, which is doubtless an advantage, but this is the core idea of plasma polymerization. See historical texts, for example, H. Yasuda. Plasma Polymerization, 1st edition, 1985 (https://www.sciencedirect.com/book/9780127687605/plasma-polymerization).

Finally, there is also one point I would like to remark. The authors do mention in different parts of the text, that plasma can penetrate porous substrates. I will recommend clarifying that atmospheric-pressure plasmas are able for that; this is not the case, for example, of low-pressure plasmas. Moreover, atmospheric-pressure plasmas are not always able to enter porous materials. It depends on the porous size and the conditions at which the plasma is operated. However, film-forming species created in the plasma – in the present case due to the AA polymerization – can diffuse over porous materials (see, for example, https://doi.org/10.1016/j.apsusc.2020.147979).

To summarize my opinions and requirements, I consider the authors have properly improved the quality of the manuscript, but I would insist on the conceptualization from a "plasma polymerization" point of view.

Author Response

  1. I find that the conceptualization of the study should be improved from a plasma point of view. Polymerization of Acrylic Acid (AA) using non-thermal plasmas is not a novel technique. The authors removed a reference they cited in the previous version (https://doi.org/10.1002/ppap.200800089) that could be valuable to be cited. As it is mentioned in this revisited article, plasma polymerization of AA has been studied for several decades and some comments about that could be included. Throughout the manuscript, the authors insist on the fact that the polymerization is carried out without additives or solvents. I completely understand and support this idea, which is doubtless an advantage, but this is the core idea of plasma polymerization. See historical texts, for example, H. Yasuda. Plasma Polymerization, 1st edition, 1985 (https://www.sciencedirect.com/book/9780127687605/plasma-polymerization).

Response: The article mentioned above has been included in the revisions made. The abstract, introduction and conclusion have been updated to include the more specific novelty of our paper which is bulk polymerization from the liquid state of AA. All previous works use more complex setups to produce PAA depositions onto substrates from AA in the gaseous state. We show bulk polymerization from the liquid state which is beneficial for impregnation of mesoporous substrates for material modification. We also expanded on the previous works using plasma for the polymerization of AA and specified the distinction from those works and ours (last two paragraphs of intro). The only reason we mentioned the fact that plasma polymerization does not require chemical additives as an advantage is to distinguish it as a preferred method to the still more widely used photo and chemical polymerization techniques. Many readers without a background in plasma may not know of this advantageous characteristic. We think it is an important aspect to emphasize for plasma polymerization to become more widely used.

  1. Finally, there is also one point I would like to remark. The authors do mention in different parts of the text, that plasma can penetrate porous substrates. I will recommend clarifying that atmospheric-pressure plasmas are able for that; this is not the case, for example, of low-pressure plasmas. Moreover, atmospheric-pressure plasmas are not always able to enter porous materials. It depends on the porous size and the conditions at which the plasma is operated. However, film-forming species created in the plasma – in the present case due to the AA polymerization – can diffuse over porous materials (see, for example, https://doi.org/10.1016/j.apsusc.2020.147979).

Response: We have taken your suggestion and specified atmospheric-pressure plasma in the work we have conducted as having the capabilities of penetrating mesoporous material. We also referenced the paper above where we discussed that not all plasmas (low-pressure plasma) have this ability to penetrate within 1-2 mm thick substrates. 

Reviewer 2 Report (Previous Reviewer 3)

After the revision of the answers and the new version of the manuscript, I do not have further comments, since each answer was addressed

Author Response

Thanks for your efforts. 

Reviewer 3 Report (New Reviewer)

This manuscript deals with another type of AP-DBD plasma application using the acrylic acid plasma-assisted polymerization. The results have very nice practical background and some suggested changes should help to increase the significance of this work:

1) As mentioned above, this work has clear practical application, so I suggest that the authors analyze the cost efficiency breakdown for such process. For example they can evaluate the OPEX and CAPEX on the basis of the open source data for the plasma setups and precursors of technical grade, electricity, etc. There are numerous TEA / LCA analytical papers dealing with plasma processes as well. Please provide this section to strenghen your work. Please compare your cost efficiency with available solutions for waterproof papers. 

2) Please provide a scheme of your plasma setup and the process. I believe that readers of Polymers will appreciate.

3) Please provide photos of your materials before and after testing, with and without plasma coatings. I believe this will demonstrate the effect clearly.

4) In your References You have talked about AA plasma polymerization processes mainly. However, you may also show CO2/C2H4 plasma polymerization, Maleic Anhydride, Maleic ANhydride-VTMOS atmospheric plasma processes:

  • 10.1016/j.surfcoat.2015.11.039
  •  
  10.1021/cm011139r 10.1016/S0257-8972(02)00407-3  

English is fine

Author Response

  • As mentioned above, this work has clear practical application, so I suggest that the authors analyze the cost efficiency breakdown for such process. For example they can evaluate the OPEX and CAPEX on the basis of the open source data for the plasma setups and precursors of technical grade, electricity, etc. There are numerous TEA / LCA analytical papers dealing with plasma processes as well. Please provide this section to strengthen your work. Please compare your cost efficiency with available solutions for waterproof papers. 

Response: This is a great suggestion. We have chosen to continue to work on this and submit the results to another MPDI journal, such as Plasma. we are unable to finish this work in a short time as requested by this journal, which is a focus on polymer anyway. 

  • Please provide a scheme of your plasma setup and the process. I believe that readers of Polymers will appreciate.

We included a scheme of the device in figure 1.

  • Please provide photos of your materials before and after testing, with and without plasma coatings. I believe this will demonstrate the effect clearly.

Response: We omitted photos of the substrates before and after polymerization because they look identical in both cases as the polymer does not yellow or char after plasma treatment. 

  • In your References You have talked about AA plasma polymerization processes mainly. However, you may also show CO2/C2H4 plasma polymerization, Maleic Anhydride, Maleic ANhydride-VTMOS atmospheric plasma processes:

Response: The goal of this paper was simply to demonstrate AA polymerization from the liquid state using DBD plasma as all previous works are accomplished from the gaseous state. In future work, we will try different liquid monomers to expand on this and their potential applications, and the work will be published in a plasma journal.

Reviewer 4 Report (New Reviewer)

the article is clear and well written however some minor remarks attached.
the physical aspect of plasmas is not sufficiently developed a parametric study according to the conditions of discharges would be interesting: power, pulse size, current, voltage, etc...

Author Response

Response: This kind of work has already been studied in previous works for the polymerization of gaseous AA (several citations in the text, 15, 17-19). However, since our work is uniquely done for AA in the liquid phase there may be some drastic variations in the results seen. Therefore, we have added to the last sentence of our conclusion that we plan on conducting this parametric study in future work. The amount of data we can compile for this work is quite vast and can be done for several liquid monomers in a comparative systematic study versus gas phase DBD polymerization.

Round 2

Reviewer 3 Report (New Reviewer)

The anwers from Authors are reasonable. 2 minor suggestions:

1) Fig 3B please remove the unnessary data table. You may leave the equation with bigger font.

2) References. You may increase the number of references by expanding the methods ysed fir COOH functionalization. Please look to my last comment from first round of revision. Right now the inyroduction is rather short.

Author Response

Here is the updated version with the figure edited as the reviewer requested. About the second comment "depositing COOH groups using different monomers", we didn't make the changes since we are not sure how to mingle that into the introduction. We also thank the reviewer that this change can be optional so we decide to send in the current version without delay. thanks.  

This manuscript is a resubmission of an earlier submission. The following is a list of the peer review reports and author responses from that submission.

Round 1

Reviewer 1 Report

The manuscript describes in situ polymerization of Acrylic Acid with dielectric barrier discharges. However the paper is quite confused and the novelty not well explained against the current status of the research. The discussion of the method and of the experiments is lacking essential details. So it is difficult to assess whether the results are significant.

Here are some points that should be clarified by the authors:

-                  In title the authors says “DBA Plasma” and in all article they talking about DBD discharge;

-       The experiments and materials are very briefly treated. Even if the installation was described in other articles, the discharge parameters should be described, variable parameters,

-       In results and discussion section, the Figure 1 must change, the modification of FTIR peaks are not discernible, also in the discussion the FTIR peaks were only identified, the difference between Acrylic Acid monomer and polymerized Acrylic Acid was not discussed.

-       To demonstrate the polymerization of AA inside the pores, additional determinations are needed (SEM images, for example, can demonstrate this). Just discussing FTIR spectra is not enough.

-       The description of the Figure 5 is not clear.

Reviewer 2 Report

The study provided by the authors is dealt with the polymerization of acrylic acid (AA) using a DBD plasma source. They used not only flat substrates but also mesoporous paper to deposit the polyacrylic acid (PAA) thin film. FTIR in ATR mode is used as characterization technique to discuss the polymerization of AA. However, no more characterization techniques have been applied, making the study misleading in the context of materials science & polymer studies, with many different techniques usually used (XPS, contact angle, profilometry, etc.)

I think that the point about "bulk polymerization" is not enough novel to justify the novelty of the study in the context of plasma polymerization. Moreover, the application proposed (wet wipe) is not original, taking into account that the water absorption of PAA coatings is perfectly known by the community .

The provided data are not discussed according to the extensive literature about plasma and acrylic acid polymerization. For example, the authors cited the revisited article by D. Hegemann and collaborators (reference 9 in the manuscript), but just in a general sentence about plasma (page 1, line 37) and not according to the discussed kinetics (rest of the manuscript).

The experimental set-up is not explained, something that I cannot understand taking into account that the article is very short. Just four lines are not enough to comprehensive understand the process (see page 2, lines 56-59). Furthermore, the reference cited in line 58 should be wrong, because it is not an article from the authors nor explaining a plasma source.

I also find questionable how the monomer is uniformly spread on the substrates to be sure about the reproducibility of the results. This point should be better explain to understand and reproduce the study.

I highly recommend the authors improving the quality of figure 2, indicating right as (a) and left as (b), explaining in (b) what is the red line and what the black (I suppose the fitting the former and experimental data the latter). I think that is also important to probe in this plot that the 1615 cm-1 band is constant.

Finally, I do not understand the last part of the discussion about contact or non-contact plasma polymerization, because I cannot properly interpret the pictures in figure 5 and also the text in page 6, lines 144-156.

Reviewer 3 Report

·      The paper deals with an interesting topic for practical application in the field of polymerization of acrylic acid. In general, the purpose and novelty of the paper are well described. 

·      The title corresponds to the purpose of the article but the use of an abbreviation (i.e. DBA) restricts the attention of the paper.

·      The abstract is concise and with the appropriate information.

·      For the introduction, there are other works related to DBD for acrylic polymerization, which should be included. 

·      Experimental details are not completely clear. 

·      In the experiments and materials section. Please provide the MW of the acrylic acid used. Also, it is mentioned that a drop of the monomer was uniformly spread, how it was done? Additionally, there is no description of the plasma generation device in the reference provided. Clarify for better understanding and provide the proper information. For the FTIR characterization, it is mentioned that the samples were analyzed before and after polymerization. Explain how the authors managed the polymerization before plasma polymerization and without solvents. We must assume it was liquid (or semi), how this state affected FTIR characterization? For the wet wipe application, provide information about the paper used. This is relevant information since the kind of paper changes in properties. 

·      In the results and discussion section, sometimes the word Figure X, appears in bold and sometimes not. Thus, do the proper correction. 

·      Improve the quality of Figure 5.

·      In the abstract, it is mentioned that the mechanism for the polymerization is demonstrated; however, in the results and discussion section, it will be explained the chemical mechanism. There are few lines mentioning that it is a first-order reaction and the reaction depends on the monomer and radical concentration. But it is important to provide the reaction mechanism for in-situ polymerization that correlates with the findings by FTIR. Further, what will be the difference between polymerization in the case of gold-coated substrate and polymerization on the paper substrate? What is the role of cellulose fibers in this case? Please provide the FTIR of the paper used as substrate.

·      The authors claim that non-contact polymerization inside a tube of a paper substrate occurs and the polymerization of AA is the same. Please provide the proper spectra for this result. How the data were treated since the substrate is paper?

·      Show clear evidence that bulk polymerization was done in a mesoporous material.

·      Strengthened of the conclusion is required in order to highlight the importance of the results, particularly when it is mentioned “since plasma is able to penetrate porous substrates, DBD plasma polymerization is a favored method for the bulk polymerization in a porous environment…”